# Dysregulated Expression of Three Genes in Colorectal Cancer Stratifies Patients into Three Risk Groups

**DOI:** 10.3390/cancers14174076

**Published:** 2022-08-23

**Authors:** Alba Rodriguez, Luís Antonio Corchete, José Antonio Alcazar, Juan Carlos Montero, Marta Rodriguez, Luis Miguel Chinchilla-Tábora, Rosario Vidal Tocino, Carlos Moyano, Saray Muñoz-Bravo, José María Sayagués, Mar Abad

**Affiliations:** 1Department of Pathology and IBSAL, University Hospital of Salamanca, 37007 Salamanca, Spain; 2Cancer Research Center and Hematology Service, University Hospital of Salamanca, 37007 Salamanca, Spain; 3General and Gastrointestinal Surgery Service, University Hospital of Salamanca, 37007 Salamanca, Spain; 4Medical Oncology Service and IBSAL, University Hospital of Salamanca, 37007 Salamanca, Spain; 5Clinical Biochemistry Service, University Hospital of Salamanca, 37007 Salamanca, Spain

**Keywords:** colorectal cancer, tumour aggressiveness, GEP, biomarkers

## Abstract

**Simple Summary:**

The main prognostic factor of sporadic colorectal cancer (sCRC) is marked by the metastatic spread of the primary tumour. Although notable progress has been made in the study of the molecular processes involved, the genomic profile responsible for the aggressiveness of the tumour process has not yet been precisely defined. Based on previous gene expression studies, we simultaneously analysed the expression profile of 28 genes, previously associated with tumour aggressiveness and/or metastatic processes, together with a variety of clinical–biological and histopathological characteristics of the disease. This study was carried out with a total of 66 consecutive patients with sCRC with the objective of establishing a prognostic scoring system based on the altered expression of those transcripts that influence overall survival (OS). Here, we show how the altered expression of the *ADH1B*, *BST2*, and *FER1L4* genes allows patients to be stratified into three risk groups that are directly associated with different 5-year survival rates.

**Abstract:**

Despite advances in recent years in the study of the molecular profile of sporadic colorectal cancer (sCRC), the specific genetic events that lead to increased aggressiveness or the development of the metastatic process of tumours are not yet clear. In previous studies of the gene expression profile (GEP) using a high-density array (50,000 genes and 6000 miRNAs in a single assay) in sCRC tumours, we identified a 28-gene signature that was found to be associated with an adverse prognostic value for predicting patient survival. Here, we analyse the differential expression of these 28 genes for their possible association with tumour local aggressiveness and metastatic processes in 66 consecutive sCRC patients, followed for >5 years, using the NanoString nCounter platform. The global transcription profile (expression levels of the 28 genes studied simultaneously) allowed us to discriminate between sCRC tumours and nontumoral colonic tissues. Analysis of the biological and functional significance of the dysregulated GEPs observed in our sCRC tumours revealed 31 significantly altered canonical pathways. Among the most commonly altered pathways, we observed the increased expression of genes involved in signalling pathways and cellular processes, such as the PI3K-Akt pathway, the interaction with the extracellular matrix (ECM), and other functions related to cell signalling processes (*SRPX2*). From a prognostic viewpoint, the altered expression of *BST2* and *SRPX2* genes were the only independent variables predicting for disease-free survival (DFS). In addition to the pT stage at diagnosis, dysregulated transcripts of *ADH1B*, *BST2*, and *FER1L4* genes showed a prognostic impact on OS in the multivariate analysis. Based on the altered expression of these three genes, a scoring system was built to stratify patients into low-, intermediate-, and high-risk groups with significantly different 5-year OS rates: 91%, 83%, and 52%, respectively. The prognostic impact was validated in two independent series of sCRC patients from the public GEO database (n = 562 patients). In summary, we show a strong association between the altered expression of three genes and the clinical outcome of sCRC patients, making them potential markers of suitability for adjuvant therapy after complete tumour resection. Additional prospective studies in larger series of patients are required to confirm the clinical utility of the newly identified biomarkers because the number of patients analysed remains small.

## 1. Introduction

Sporadic colorectal cancer (sCRC) can be defined as a biologically very heterogeneous disease in terms of its clinical manifestations, molecular characteristics, therapeutic response, and patient prognosis [1,2], and it continues to be one of the most frequent causes of cancer death in the western world. Most deaths, of up to a third of patients with sCRC, are caused by the metastatic spread of the primary tumour that can occur between the time of diagnosis and two years after surgery on the primary tumour [3]. The appearance of these metastases is usually preceded by the appearance of positive lymph nodes and/or lymphovascular invasion, which are currently the main clinical parameters used to predict the evolution of these tumours [4]. We [5,6,7] and others [2,8,9] have recently shown that sCRC metastasis may emerge in the context of a specific genetic tumour background associated or not with other genetic alterations, and that this further affects cellular control of growth and proliferation [8]. The discovery of those specific genetic alterations that might contribute towards identifying patients with locally aggressive tumours or who are at risk of developing metastases could significantly influence the development of new strategies for the diagnosis and management of the disease.

In recent years, methods of microarray analysis, such as gene expression profiling, have allowed the simultaneous study of several thousand cancer-specific genes [5]. Gene expression profiling is an interesting tool for identifying new biomarkers associated with prognosis and treatment in various types of neoplasia, including sCRC [8]. In breast cancer, GEP studies are used as a screening tool to identify molecules to be targeted by existing or future (customized) therapies, as well as predictors of early relapse, such as the PAM50 (Prosigna; NanoString Technologies, Seattle, WA, USA) and Oncotype DX Recurrence Score (RS; Genomic Health, Redwood City, CA, USA), among other profiling platforms. To identify patients for whom the potential benefits of chemotherapy outweigh the risks, and who could thereby avoid early relapses, several GEP studies have been postulated as predictors of the evolution of patients with stage II sCRC, such as the Oncotype DX^®^ colon cancer test (Genomic Health, Inc., Redwood City, CA, USA) [10] and Coloprint^®^ gene chips (Agendia, Inc., Irvine, CA, USA) [11,12]. However, the poor prognosis of a patient with sCRC is usually due to aggressive locoregional tumour growth or metastatic spread of the primary tumour, mainly into the liver. Although sCRC is amongst the best characterized solid tumours at the molecular level, the specific genes and molecular mechanisms determining the aggressiveness of the tumour remain to be exhaustively identified. For this reason, we [6,7] and others [13] have focused on molecular analysis to study the primary tumour and their paired liver metastases in depth. In all these studies, GEP abnormalities have been investigated, as well as altered signalling pathways or miRNAs that regulate the expression of the genes involved in the process of tumour progression. Overall, the metastatic tumour samples studied displayed a GEP that was highly similar to that of their paired primary tumour. However, we found 52 mRNAs and two miRNAs to be differentially expressed between the two tumour tissues analysed [6,7]. In addition, by assessing the GEP array of coding and noncoding genes in metastatic vs. nonmetastatic primary sCRC tumours followed for >5 years, Gutiérrez et al. identified 14 mRNAs and five miRNAs associated with the aggressiveness of the sCRC, all of which might play a role in the development of sCRC metastasis, [7]. However, an in-depth study of this GEP in an independent series of consecutive patients is needed to establish correlations with the clinical–biological behaviour of the disease, with special emphasis on those characteristics involved in the pathogenesis and prognosis of the disease, as well as those related to the metastatic process.

Here, we evaluate the differential expression of 28 genes previously selected from a high-density array (n = 49,395 mRNAs in a single assay) for their possible association with tumour local aggressiveness and metastatic processes in 66 consecutive patients with sCRC. Our main objective is to identify GEPs that help us detect, at the time of diagnosis, patients with aggressive tumours after surgical resection of the primary tumour, in order to avoid local or distant relapses.

## 2. Material and Methods

*Patients and samples.* Tissue specimens from 66 consecutive sporadic colorectal cancer (sCRC) patients, after undergoing surgery for resection of the primary tumour and before any cytotoxic treatment (between January and December 2015), were analysed. All of them were stored in OCT at −80 °C in the Tumor Biobank of the University Hospital of Salamanca, Red de Bancos de Tumores de Castilla y León, Salamanca, Spain. Informed consent was given by all patients prior to their entering the study, in accordance with the Declaration of Helsinki. The study was approved by the local ethics committee of the University Hospital of Salamanca (Salamanca, Spain). In all cases, tumours were diagnosed and classified according to the AJCC (American Joint Committee on Cancer) criteria [14] by a pathologist from the Pathology Department of the University Hospital of Salamanca. The median follow-up at the time the study was completed was 65 months (range: 5–79 months). Five of the patients showed synchronous liver metastases (8%); by tumour grade, 18 cases were classified as well-differentiated, 46 were moderately differentiated, 1 was poorly differentiated, and 1 was an undifferentiated carcinoma. In all cases, histopathological grade was confirmed in a second, independent evaluation by an experienced pathologist. In parallel, normal colonic tissue samples were taken at a minimum distance of 10 cm from the tumour site of 10 of the 66 patients included in the present study.

Primary tumours were localized in the rectum (n = 33), or in the right (caecum, ascending or transverse; n = 26) or the left (descending and sigmoid; n = 7) colon. The median size of the primary tumours was 4.0 ± 2.6 cm, with the following distribution according to their TNM stage at diagnosis [14,15]: stage I, 14 cases; stage II, 21 cases; stage III, 26 cases; and stage IV, 5 cases (Table 1).

*RNA extraction and GEP studies.* Samples of paraffin-embedded tissue (FFPE) were deparaffinized by three washes in Histoclear II (3 min at 50 °C). They were then centrifuged for 2 min at maximum speed and the supernatant was discarded. After washing in absolute ethanol twice (maximum speed for 2 min), the deposit was allowed to dry at room temperature for 10–15 min. Digestion buffer containing 20 mg/mL protease K (Thermo Fisher Scientific Inc., Waltham, MA, USA) was added to each sample and allowed to lyse at 55 °C overnight. The next day, Easy-BLUETM (iNtRON) was used to dissociate the nucleoprotein complexes; subsequently, chloroform was added, and the tubes were shaken vigorously for 15 s. The samples were centrifuged at 12,000× *g* for 15 min at 4 °C and the aqueous phase was transferred to a new tube. Total RNA was precipitated by mixing with isopropyl alcohol; once obtained, it was placed at −20 °C for at least 1 h and subsequently centrifuged at maximum for 10 min at 4 °C. The RNA pellet was dissolved in 200 µL DEPC water (Ambion, Austin, TX, USA) and precipitated with cold 100% ethanol, 10% sodium acetate, pH 5.2 and 2 µL glycogen 20 mg/mL (Thermo Fisher Scientific) at −20 °C for at least 3 h. Subsequently, the sample was centrifuged for up to 10 min at 4 °C and washed with 70% ethanol on ice. Finally, the RNA pellet was dissolved in 10 µL DEPC water.

The Nanodrop platform was used to verify the quality and quantity of the extracted RNA. The total volume of isolated RNA was treated with DNAseI Amplification Grade (Sigma, St. Louis, MO, USA), and the DNA-free RNA was quantified using a Quant-iT RNA Assay kit (Thermo Fisher Scientific), following the manufacturer’s instructions. For GEP studies, the NanoString nCounter system was used, which assigns a unique colour code to each of the genes under study for subsequent reading by fluorescence. Starting with the isolated RNA, all initial concentrations were adjusted to work with a final RNA concentration of 60 ng/µL. After making the necessary dilutions, the hybridisation reaction was prepared, in which the RNA binds to the capture probe and the labelling probe by incubation at 65 °C for about 18 h. Once the hybridisation reaction had been completed, the samples were passed to the nCounter Prep Station, where the unbound probe residues were removed, and the labelled sequences were immobilized in the corresponding cartridge. The cartridge was then immediately read in the Digital Analyzer, a device for reading the fluorescent labelling of each of the genes of interest. The primer sequencers used for each gene are shown in Appendix A. An RCC file of the level of expression of each of the genes was then generated from the reading data of the genes. In addition to the genes of interest, three control genes (*GAPDH*, *ACTB*, *TUBB*) were included in each experiment.

Raw data were preprocessed and quality-checked using the nanoR package (version 0.1.0) in R (version 4.0.4) (Viena, Austria). Counts were then normalized through the housekeeping method based on the expression of the *TUBB*, *ACTB*, and *GAPDH* genes and log_2_-transformed. Unsupervised multidimensional scaling (MDS) was performed and dendrograms produced using the SIMFIT statistical application (version 7.5.4, Universidad de Salamanca (Salamanca, Spain)), taking the Euclidean distance as the distance measure and the group average as the linkage method. Based on the MDS results, we determined the presence of two or three CRC subgroups by applying an unsupervised learning Gaussian mixture model in the mclust package (version 5.4.8) in R. Differential expression analysis was carried out with the limma package (version 3.46.0). Genes that attained a Benjamini–Hochberg adjusted value of *p* < 0.05 were considered significant. Gene overrepresentation was analysed on the WebGestalt server (PMID: 31114916) using KEGG and Reactome sources for pathway analysis. Gene Ontology biological process and molecular function analysis were used as the source for gene function analysis. Prognostic scores were calculated based on the expression of the *ADH1B* (NM_001286650), *BST2*, and *FER1L4* genes. The cut-off levels for defining low and high risk were: *ADH1B* (NM_001286650): <4.39 for low risk (LR) and ≥4.39 for high risk (HR); *BST2* gene: <10.34 for HR and ≥10.34 LR; *FER1L4* pseudogene: <4.73 for HR and ≥4.73 LR. For categorical variables, the χ^2^ test was used to evaluate the statistical significance of differences between groups using IBM SPSS 25.0 Statistics (IBM Corp., Armonk, NY, USA). Overall survival (OS) was established using Kaplan–Meier curves and developed with R. Univariate survival was analysed with the R survival package (version 3.2-7) using the Kaplan–Meier estimator, and curves were compared using the log-rank test. We used the Cut-off Finder application (http://molpath.charite.de/cutoff (accessed on 20 October 2021)) to determine the optimal cut-off associated with survival for each gene. This cut-off was defined as the most significant split that discriminates between long and short survival of all possible cut-offs tested using the log-rank test. Multivariate survival was also analysed in R using Cox proportional hazards regression. Variable inflation and proportionality of hazards were previously checked through the vif function in the car package (version 3.0-12) and the Schoenfeld test using the cox.zph function in the survival package, respectively. Cox models were graphically represented using forest plots with the survminer package (version 0.4.9). Our results were validated in the GSE39582 and GSE87211 series downloaded from the GEO repository.

## 3. Results

*GEP of sCRC tumours.* Supervised analysis of the sCRC GEP showed 11 mRNA dysregulated genes of a total of 28 genes analysed (previously selected from a high-density array; 49,395 mRNAs in a single assay). Most of these mRNA transcripts were upregulated in sCRC samples (9/11; 82%), while only two (18%) were downregulated. The global transcription profile (expression levels of the 28 genes studied simultaneously) allowed us to discriminate between sCRC tumours and nontumoral colonic tissues (Figure 1 and Appendix A), it being possible to classify them accurately according to this expression profile. The mRNA transcripts revealed SALL4, SPP1, THBS2, CXCL3, and SRPX2 to be the other dysregulated genes overexpressed at the highest levels, while ADH1B and MYLK were the most strongly downregulated across all sCRC samples analysed (Table 2).

The tumour-associated genes most strongly overexpressed in CCR primary tumours included those involved in cell adhesion, the extracellular matrix (ECM), and signalling processes (e.g., *SPP1* and *THBS2*), and others associated with tumour cell progression, migration, and angiogenesis (e.g., *SRPX2*, *SPP1*, and the pseudogene *FER1L4*). The oncogene *SALL4*, the proinflammatory cytokine ligands (*CXCL3*), and other genes such as *SERPINA1*, *MOCOS*, and *BST2* also had notably high levels of expression. In contrast, two genes, *ADH1B* and *MYLK*, with an established relationship with sCRC, had high levels of downregulation of tumour suppressor mRNAs. Two isoforms, with similar levels of expression, were detected in both genes (Table 2).

*Functional characterization of dysregulated GEP in sCRC tumours.* Analysis of the biological and functional significance of the dysregulated GEPs in our sCRC tumours revealed 31 canonical pathways that were significantly altered relative to nontumoral colonic tissues (Figure 2). Among the most commonly altered pathways in the sCRC tumours, we observed increased expression of genes involved signalling pathways and cellular processes such as the PI3K-Akt pathway, the interaction with extracellular matrix (ECM) or the apelin pathway (*SPP1* and *THBS2*), the heterochromatin protein formation pathways (*SALL4*), and other functions related to cell-signalling processes (*SRPX2*). It is also worth highlighting the participation of *THBS2* in cell communication functions, such as the formation of the phagosome or extracellular matrix molecules, as well as in focal adhesion processes. This group also includes genes related to immunological functions, such as the *SERPINA1* gene, which is involved in the complement pathway and in the coagulation cascade, cytokine, and interleukin signalling pathways, and the NOD-like receptor pathway (*CXCL3*), as well as others whose fundamental role is to protect against viral infections, as in the case of the *BST2* gene.

In addition, sCRC tumours also featured the downregulation of genes involved in retinol metabolism or other processes such as fatty acid degradation, glycolysis, gluconeogenesis, and drug metabolism through cytochrome P450, as in the case of the *ADH1B* gene. The other notable member of this group of downregulated genes was *MYLK*, which is directly involved in cell signalling processes such as the calcium pathway or the cGMP-PKG pathway, as well as being present in other cell communication pathways such as that associated with regulating the cytoskeleton and focal adhesion.

*Association between sCRC tumour-specific GEP and other disease features.* Table 3 shows the association between the clinical–biological and histopathological characteristics of patients analysed and those transcripts indicating stronger dysregulation. Thus, significantly decreased expression of *SALL4* and *SRPX2* genes was detected in larger (>4 cm) tumours (*p* = 0.03; *p* = 0.02, respectively), while the overexpression of *MYLK* was more frequent in patients with tumours in the rectum (*p* = 0.003), at advanced stages (*p* = 0.01; pT3–pT4), and for which metastatic lymph nodes were present (*p* = 0.05; pN1–pN2). Similarly, whereas *THBS2* overexpression was associated with tumours located in the rectum, downregulated expression levels of ADH1B mRNA were more frequent in tumours situated in the colon (*p* = 0.02) at advanced stages (T3–T4; *p* = 0.04).

*Impact of GEP and other disease features on patient disease-free survival (DFS) and overall survival (OS).* From a prognostic viewpoint, two genes with altered mRNA in our genomic signature were confirmed in the univariate analysis as markers related to lower DFS: *ADH1B* (NM_001286650; *p* = 0.01) and *SALL4* (*p* = 0.02). As expected, the presence of lymphovascular invasion was also significantly associated with a higher incidence of relapses and a shorter DFS (*p* = 0.02; Figure 3).

Regarding OS, four mRNA dysregulated genes, *ADH1B* (NM_001286650) (*p* = 0.03), *MYLK* (NM_053026) (*p* = 0.01), *BST2* (*p* = 0.05), and *FER1L4* (*p* = 0.04), were the only individual parameters associated with patient outcome. Thus, higher levels of expression of the *ADH1B* (NM_001286650) and *MYLK* (NM_053026) genes and/or lower expression levels of the *BST2* and *FER1L4* genes were all associated with significantly shorter OS (Figure 4). Multivariate analysis of prognostic factors showed that the altered expression of *BST2* and *SRPX2* genes were the only independent variables predicting DFS (Figure 5, panel A). In addition to the pT stage at diagnosis, dysregulated transcripts of *ADH1B* (both isoforms, NM_001286650 and NM_000668), *BST2*, and *FER1L4* genes showed a prognostic influence on OS in the multivariate analysis (Figure 5, Panel B). Based on the altered expression of these three genes, a scoring system was developed to stratify patients into low-risk (transcripts for *ADH1B*-NM_001286650-, *BST2*, and *FER1L4* expressed at low levels: score 0; n = 11), intermediate-risk (one transcript expressed at high levels: score 1; n = 23), and high-risk (two or three transcripts expressed at high levels: score 2; n = 21) groups with significantly different (*p* = 0.002) 5-year OS of 91%, 83%, and 52%, respectively (Figure 6).

*Validation of the prognostic score in two independent series of sCRC patients.* In order to confirm the prognostic impact of the altered expression of significant genes for OS in the multivariate analysis, we investigated its prognostic impact in two independent series of sCRC patients from the public GEO database (n = 566 patients; survival information was available from 562 cases and n = 203 cases; survival information was available from 196 cases). Dysregulated expression of *ADH1B* and *BST2* genes was also found to have poorer OS in the first series. These results confirm the prognostic impact of dysregulated transcripts of *ADH1B* (NM_001286650) and *BST2* genes. However, the prognostic impact of the level of expression of the *FER1L4* gene could not be confirmed in this independent sCRC series of patients because the information was not available (Appendix A). In contrast, in the second series, the prognostic impact of the altered expression of the *FER1L4* is confirmed. Nevertheless, information for the *BST2* and *ADH1B* genes was not available in this database to validate the prognostic impact of altered expression of these two genes (Appendix A).

## 4. Discussion

Sporadic colorectal cancer (sCRC) is currently the second most deadly cancer worldwide, so there is a pressing need to gain a deeper understanding of the molecular and genetic events behind the adverse prognosis of the disease. The American Joint Committee on Cancer (AJCC) tumour–node–metastasis (TNM) staging system is the most widely used prognostic standard for sCRC. It has been revised several times over the years to improve its prognostic performance and the treatment suggestions for patients with sCRC [14,15]. However, histologically identical tumours, of the same stage, may exhibit completely different clinical behaviours throughout the course of the disease, making it necessary to include new biomarkers that allow us to identify patients with an adverse prognosis and a higher risk of relapse and/or progression. Based on our previous studies of GEP in sCRC tumours using a high-density array (50,000 genes and 6000 miRNAs in a single assay) [5,7], we selected a 28-gene signature that was found to be associated with an adverse prognostic value for predicting patient survival. Our major goal was to identify new biomarkers that, when combined, might predict the aggressiveness of the tumour, already at diagnosis, of sCRC patients.

In our series, the highest detected levels of mRNA transcripts in tumour tissue were those of the *SALL4* gene, whose altered expression was associated with tumour size and adverse prognosis. Consistent with our observations, Forghanifard et al. [16] described the association between overexpression of the *SALL4* oncogene with tumour differentiation grade and advanced stages [16,17,18,19]. It is important to highlight that it is possible to detect the mRNA transcript levels of *SALL4* in the peripheral blood (PB) of sCRC patients; thus, Khales et al. [18] described the greater diagnostic sensitivity and specificity for the detection of mRNA SALL4 compared with the classic CEA marker in PB of sCRC patients, which could facilitate initial prognostic stratification and follow-up of patients as a liquid biopsy. In general, our results showed the positive regulation of genes associated with processes related to tumour progression and metastatic process, including genes involved in extracellular matrix (ECM) interaction and cell adhesion, such as *SPP1* and *THBS2* [20,21,22,23,24,25,26,27,28]. In fact, Allan et al. [25] highlighted *SPP1* as a key player in lymphatic metastasis processes in breast cancer and showed increased expression in metastases compared with their corresponding primary tumours. Similarly, Ng et al. [29] demonstrated how the overexpression of *SPP1* activates the PI3K-AKT pathway, promoting the epithelial–mesenchymal transition and leading to the development of metastases in sCRC patients. Another key role of *SPP1* is associated with immune response [24], regulating host immunity, and being associated with increased cell proliferation by preventing tumour cell apoptosis. All these results suggest that the dysregulated transcription of *SPP1* favours the loss of cell adhesion that allows tumour cells to dissociate from the primary tumour mass, favouring tumour cell invasion and dissemination. Another strongly expressed gene found in tumour tissue was *CXCL3*, a proinflammatory cytokine ligand related to immunological processes, consistent with previous observations. Sun et al. [30] related the overexpression of *CXCL3* with a short OS in sCRC patients. Xiong et al. [31] significantly associated the high level of expression of *CXCL3* with lymphatic invasion, distant metastases, and advanced tumour stage. A recent study found these same findings in pancreatic ductal adenocarcinoma [32]. According to the findings identified by other authors, we also detected a high level of expression of the *SERPINA1* gene, a protease inhibitor, in patients with advanced stages (pT3–pT4) [33].

Positive regulation of the *FER1L4* pseudogene is well known as an oncogenic driver associated with poor prognosis in other neoplasms, such as breast carcinoma and clear cell renal carcinoma [34,35]. Interestingly, Ostovarpour et al. [36] found a correlation between expression levels of *FER1L4* and the malignant transformation of the epithelial cell in sCRC, suggesting that expression level could be considered a potential biomarker for sCRC development [36]. However, we should emphasise that there are hardly any published studies that analyse the expression of this gene with the prognosis of the disease.

In contrast, *ADH1B* and *MYLK* mRNAs showed the most pronounced downregulation in all the sCRC samples studied. The *ADH1B* gene is involved in ethanol metabolism, by-products such as acetaldehyde and reactive oxygen species, increasing toxicity, and the risk of sCRC, as previously demonstrated by Chen et al. [37]. Recently, Zhao et al. [2] described low levels of the *ADHB1* transcript commonly occurring in sCRC patients. We had previously highlighted the reduced expression of *MYLK* in liver metastases relative to normal tissue, while it was expressed at normal levels in primary tumours [5]. To the best of our knowledge, this is the first report of significantly increased expression of *MYLK* in sCRC patients associated with poor prognosis, advanced-stage tumours, and significantly short OS. Further studies are needed to determine the precise role of *MYLK* gene in sCRC patients.

From a prognostic point of view, the altered expression of the *SRPX2* and *BST2* genes was associated significantly shorter disease-free survival (DFS). However, we found few published studies that focused on the expression of these genes in CRC. In this regard, Gao et al. [38] reported elevated *SRPX2* expression levels associated with an adverse prognosis in patients with pancreatic cancer. Lin et al. [39] identified *SRPX2* expression levels as an independent prognostic marker, promoting cell migration and invasion in hepatocellular carcinoma. *SRPX2* is also believed to be associated with angiogenesis in sCRC patients [40]. These findings support those of our study, whereby *SRPX2* expression is proposed as a predictive biomarker related to prognosis in patients with sCRC. However, no studies have analysed the expression of the *BST2* gene in sCRC tumours. In addition to *BST2* expression, multivariate analysis showed that *ADH1B* and *FER1L4* expression were the most informative variables in relation to OS of sCRC patients. Interestingly, the *ADH1B* gene exhibited the most pronounced downregulation in all the sCRC samples analysed. In line with other studies, and using similar methods, Gao et al. [9] found the *ADH1B* gene to be among the top five most strongly downregulated genes in CRC. It is worth noting that *ADH1B* expression is the only factor associated with OS in both the univariate and multivariate analyses. Consistent with these observations, Zhao et al. [41] found *ADH1B*, among other genes, can predict the OS rate of sCRC patients. In addition, genetic polymorphisms involving this gene have recently been associated with an increased risk of sCRC [42]. Other independent prognostic factors associated with OS were the dysregulated expression of the *BST2* and *FER1L4* genes, but as discussed above, to the best of our knowledge there are no expression studies of these genes in sCRC. As they might constitute new biomarkers of survival, their role in CRC deserves further investigation. In the current study, we have reanalysed an independent dataset and confirmed the prognostic value of the altered expression of *ADH1B* and *BST2* genes, despite the substantial differences in the technologies used in the two studies. This strengthens the evidence for the clinical relevance of both genes. In contrast, the clinical impact of the dysregulated expression of the *FER1L4* pseudogene could not be validated in this series due to the lack of oligonucleotides targeting it.

It is important to highlight that our findings have been previously observed at the protein level in patients with sCRC. However, most of these studies were performed on BST2. Chiang et al. showed that plasma levels of BST2 could discriminate sCRC patients from healthy individuals, along with CEA levels. They suggested that plasma levels of BST2 might be a novel biomarker and prognostic factor in sCRC patients [43]. Recently, univariate and multivariate analyses by Mukai et al. showed that BST-2 expression by immunohistochemistry is an independent prognostic classifier of patients with gastric cancer. They concluded that, because BST-2 is expressed on the cell membrane, it could be a therapeutic target for oesophageal, gastric and sCRC [44]. Unlike BST2, there are hardly any studies that have demonstrated a correlation between the protein levels of IDH1B and FER1L4 and sCRC prognosis

In the present study, a prognostic scoring classifier based on the simultaneous analysis of the expression of the *ADH1B*, *BST2*, and *FER1L4* genes was identified. It is a potential marker of adjuvant therapy after complete tumour resection that could be used to detect sCRC patients who are still at high risk of disease recurrence and/or of having poor OS. If the prognostic value of this new risk stratification model is confirmed in prospective series of sCRC patients, it could identify those patients who might benefit from adjuvant treatment after tumour surgery.

## 5. Conclusions

We report a strong association between the altered expression of three genes and the clinical outcome of sCRC patients undergoing complete tumour resection, which are potential markers of adjuvant therapy after complete tumour resection. It should be noted that the study was carried out on the same expression platform on which the genomic markers of breast cancer patients are usually assessed to predict early relapses and avoid unnecessary postsurgical treatments (PAM50; nCounter; NanoString Technologies, Seattle, WA, USA); it is one of the most reproducible technologies across different laboratories. Additional prospective analyses in larger series of sCRC patients are needed to confirm the prognostic utility of the newly proposed biomarkers.

## Figures and Tables

**Figure 1 cancers-14-04076-f001:**
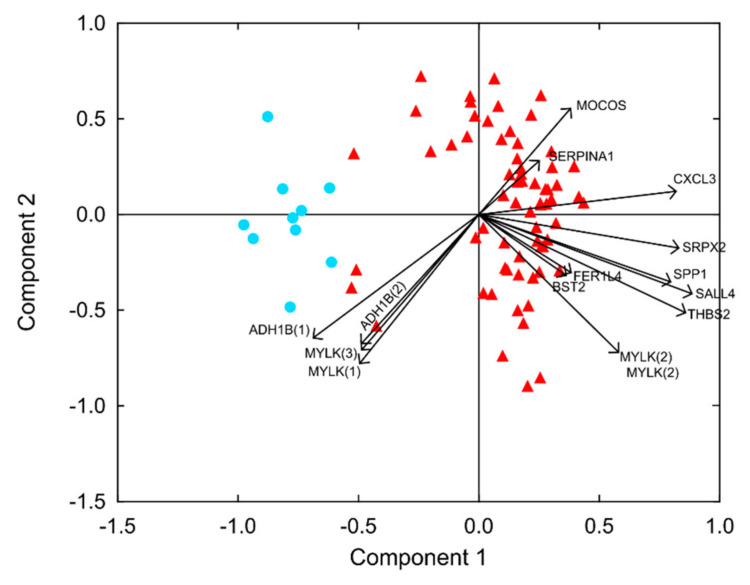
Classification of sCRC tumours and nontumoral colorectal tissues based on the gene expression profile (GEP) of the most strongly dysregulated transcripts. Biplot analysis of 66 primary colorectal tumours (red triangles) vs. 10 nontumoral colorectal tissue samples (blue circles).

**Figure 2 cancers-14-04076-f002:**
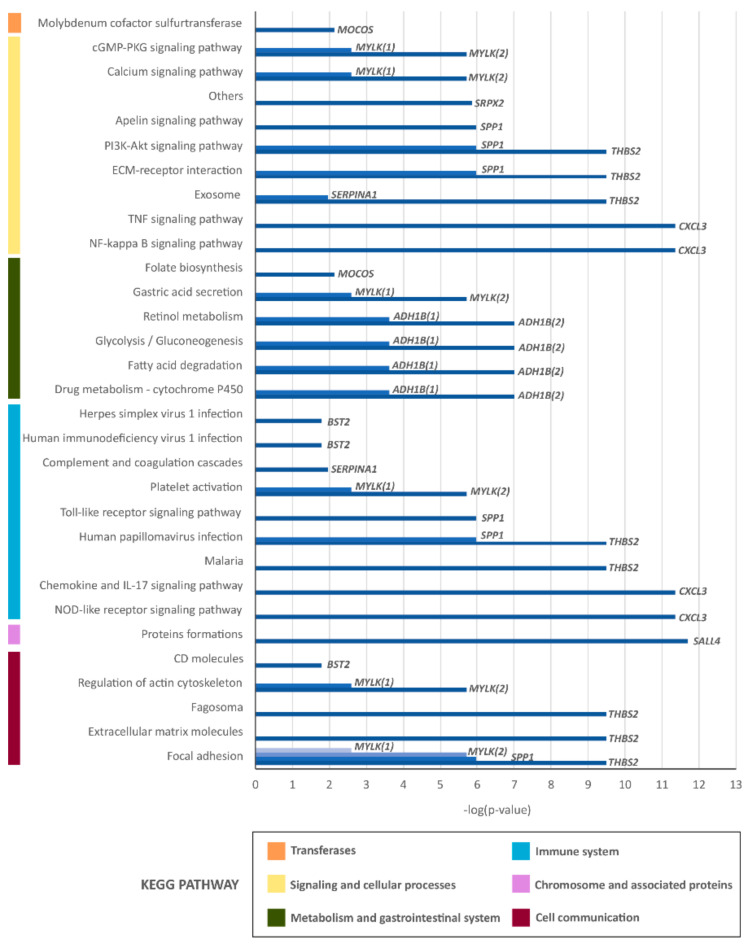
Most representative canonical pathways involved in sCRC tumours as defined by the gene expression profile (GEP) of the most strongly dysregulated transcripts.

**Figure 3 cancers-14-04076-f003:**
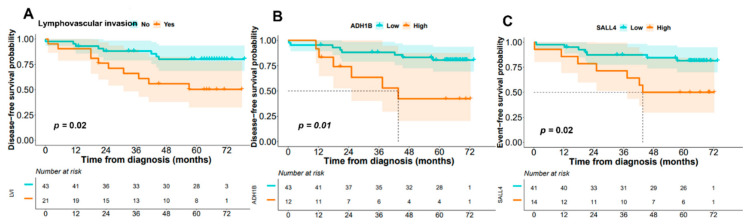
Histological and genetic features of sCRC patients showing the impact on disease-free survival (DFS) in the univariate analysis (n = 55). (**A**) Lymphovascular invasion, (**B**) *ADH1B* gene (NM_001286650), and (**C**) *SALL4* gene (NM_001318031).

**Figure 4 cancers-14-04076-f004:**
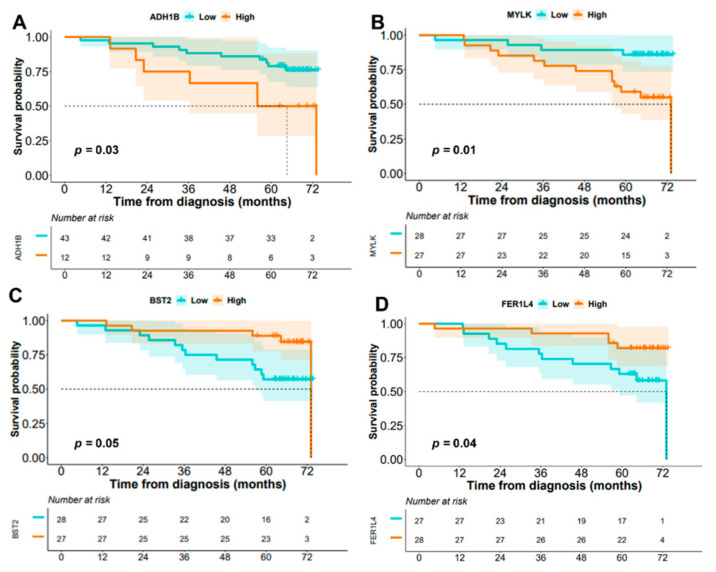
Genetic features of sCRC patients showing the impact on overall survival (OS) in the univariate analysis (n = 55). (**A**) *ADH1B* gene (NM_001286650), (**B**) *MYLK* gene (NM_053026), (**C**) *BST2* gene (NM_004335), and (**D**) *FER1L4* pseudogene (NR_119376).

**Figure 5 cancers-14-04076-f005:**
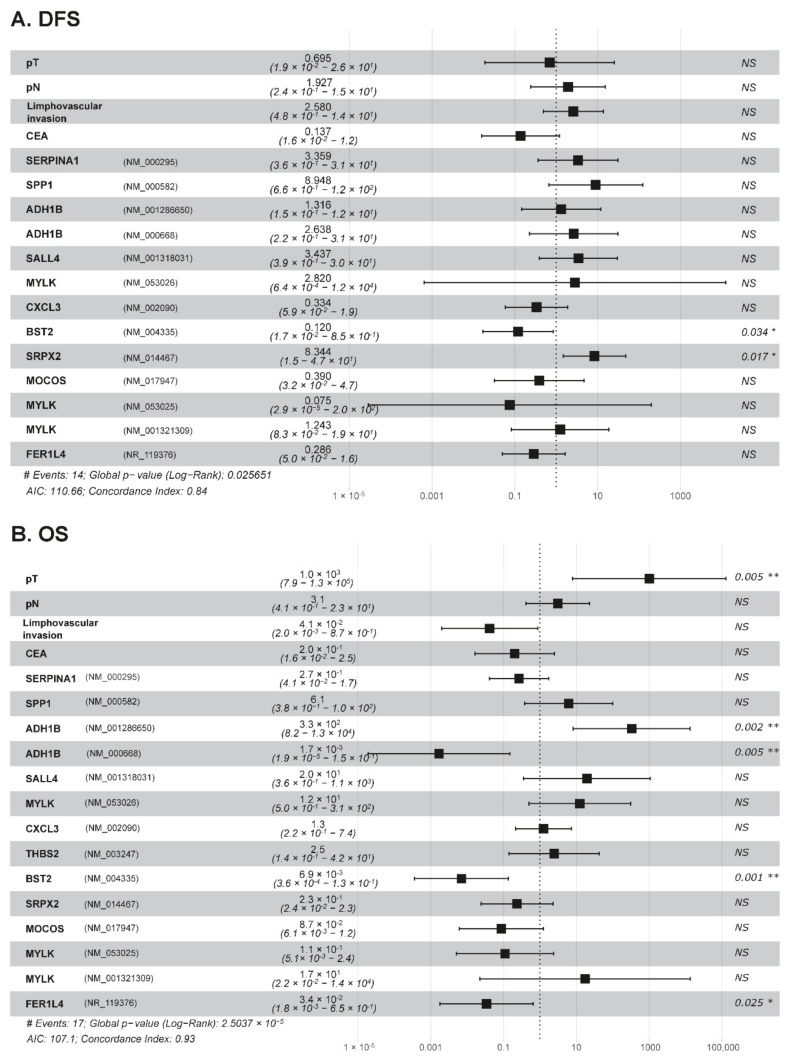
Multivariate analysis of prognostic impact of histological and genetic features. *BST2* and *SRPX2* are the only statistically significant independent predictors of disease-free survival (DFS) (panel (**A**)). Histopathological tumour classification (pT), lymphovascular invasion, *ADH1B* gene (NM_001286650), *ADH1B* gene (NM_000668), *BST2* gene, and *FER1L4* gene also showed a prognostic impact on overall survival (OS) in the multivariate analysis (panel (**B**)). Survival analysis parameters are indicated with #. CEA: Carcinoembryonic antigen. Survival information was available from 55 cases. NS: no statistically significant differences observed (*p* > 0.05). Values with “*”are those that are significant below 0.05 while those marked with “**” are less than 0.01.

**Figure 6 cancers-14-04076-f006:**
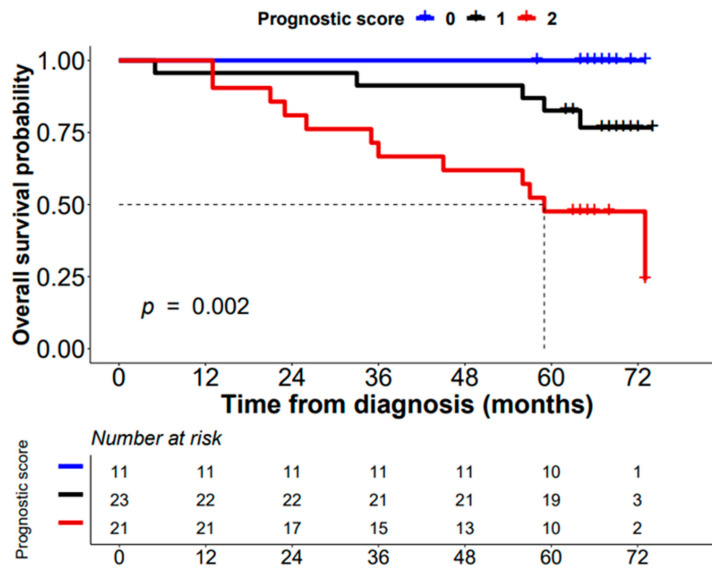
Overall survival (OS) of sCRC patients stratified by the prognostic score proposed in the present study. This prognostic score was based on the altered expression of the most relevant genes predicting OS in the multivariate analysis (*ADH1B* (NM_001286650), *BST2*, and *FER1L4*). Score 0: transcripts for *ADH1B* (NM_001286650), *BST2*, and *FER1L4* expressed at low levels. Score 1: one of these genes is expressed at a high level. Score 2: two or three of these genes are expressed at a high level.

**Table 1 cancers-14-04076-t001:** Clinical and biological characteristics of sCRC patients (n = 66) at diagnosis.

Variable	Total (%)
Age (years) *	68 (38–92)
Gender	
Female	29 (44)
Male	37 (56)
Tumour size (cm) *	4 (0.5–13)
Site of primary tumour	
Left colon	7 (10)
Right colon	26 (40)
Rectum	33 (50)
CEA serum levels (ng/mL) *	2.53 (0.54–5481)
≤5 ng/mL	54 (82)
>5 ng/mL	12 (18)
Grade of differentiation	
Well-differentiated	18 (26)
Moderately-differentiated	46 (70)
Poorly differentiated	1 (2)
Undifferentiated	1 (2)
Lymphovascular invasion	
Yes	22 (33)
No	44 (67)
Histopathological tumour classification	
pT1	5 (8)
pT2	16 (24)
pT3	39 (59)
pT4	6 (9)
Lymph node involvement	
pN0	35 (53)
pN1	15 (23)
pN2	16 (24)
Metastasis status	
M0	61 (92)
M1	5 (8)
TNM stage at diagnosis	
I	14 (21)
II	21 (32)
III	26 (39)
IV	5 (8)
Deaths	24 (36)
Overall survival (months) *	65 (5–79)

* Results are expressed as the median (range).

**Table 2 cancers-14-04076-t002:** Statistically significant mRNA dysregulated in sCRC patients (n = 66) relative to nontumoral colorectal tissues (n = 10).

Gene Name	Gene ID ^#^	Fold Change * (vs. Nontumoral)	Chromosomal Band	*p*
Transcripts upregulated in CRC relative to nontumoral tissue				
*SALL4*	NM_001318031	5.86	20q13.2	<0.001
*SPP1*	NM_000582	5.84	4q22.1	<0.001
*THBS2*	NM_003247	4.81	6q27	<0.001
*CXCL3*	NM_002090	3.94	4q13.3	<0.001
*SRPX2*	NM_014467	3.08	Xq22.1	<0.001
*SERPINA1*	NM_000295	2.65	14q32.13	0.01
*FER1L4*	NR_119376	2.56	20q11.22	0.01
*IL13RA2*	NM_000640	1.75	Xq23	0.07
*MOCOS*	NM_017947	1.73	18q12.2	0.01
*BST2*	NM_004335	0.89	19p13.11	0.02
Transcripts downregulated in CRC relative to nontumoral tissue				
*ADH1B*	NM_001286650	−5.38	4q23	<0.001
*ADH1B*	NM_000668	−4.40	4q23	<0.001
*MYLK*	NM_001321309	−2.41	3q21.1	0.003
*MYLK*	NM_053026	−2.40	3q21.1	<0.001

**^#^** NCBI reference sequence. * Fold change is expressed in logarithmic scale in base 2. Several isoforms of the *MYLK*, *SPP1*, *ITIH1*, *FBXO32*, and *ADH1B* genes were studied.

**Table 3 cancers-14-04076-t003:** Association between clinical–biological and histopathological characteristics of sCRC patients (n = 66) and the most strongly dysregulated transcripts with any significant association.

		*SALL4*			*MYLK* ^1^			*SRPX2*			*THBS2*			*ADH1B* ^2^		
*Cut-Off (Fold Change)*		<8.12	≥8.12	*p*	<7.53	≥7.53	*p*	<9.79	≥9.79	*p*	<7.60	≥7.60	*p*	<9.25	≥9.25	*p*
Tumour size (cm)	<4	18 (27)	12 (18)	0.03	11 (17)	19 (29)	NS	16 (24)	14 (21)	0.02	13 (20)	17 (26)	NS	22 (33)	8 (12)	NS
	≥4	30 (46)	6 (9)		21 (32)	15 (23)		29 (44)	7 (11)		20 (30)	16 (24)		27 (41)	9 (14)	
Site of primary tumour	Colon	27 (41)	6 (9)	NS	22 (33)	11 (17)	0.003	25 (38)	8 (12)	NS	21 (32)	12 (18)	0.05	28 (42)	5 (8)	0.02
	Rectum	21 (32)	12 (18)		10 (15)	23 (35)		20 (30)	13 (20)		12 (18)	21 (32)		21 (32)	12 (18)	
CEA serum levels (ng/mL)	≤5	40 (61)	14 (21)	NS	26 (39)	28 (42)	NS	40 (61)	14 (21)	0.03	29 (44)	25 (38)	NS	39 (59)	15 (23)	NS
	>5	8 (12)	4 (6)		6 (9)	6 (9)		5 (8)	7 (11)		4 (6)	8 (12)		10 (15)	2 (3)	
pT stage	pT1–pT2	17 (26)	4 (6)	NS	15 (23)	6 (9)	0.01	14 (21)	7 (11)	NS	14 (21)	7 (11)	NS	19 (29)	2 (3)	0.04
	pT3–pT4	31 (47)	14 (21)		17 (26)	28 (42)		31 (47)	14 (21)		19 (29)	26 (39)		30 (45)	15 (23)	
Lymph node involvement	pN0	26 (39)	9 (14)	NS	21 (32)	14 (21)	0.05	24 (36)	11 (17)	NS	21 (32)	14 (21)	NS	28 (42)	7 (11)	NS
	pN1–pN2	22 (33)	9 (14)		11 (17)	20 (30)		21 (32)	10 (15)		12 (18)	19 (29)		21 (32)	10 (15)	
TNM stage at diagnosis	I	10 (15)	4 (6)	NS	11 (17)	3 (4)	0.02	9 (14)	5 (8)	NS	10 (15)	4 (6)	NS	12 (18)	2 (3)	NS
	II	16 (24)	5 (8)		10 (15)	11 (17)		16 (24)	5 (8)		10 (15)	11 (17)		17 (26)	4 (6)	
	III	18 (27)	8 (12)		9 (14)	17 (26)		16 (24)	10 (15)		11 (17)	15 (23)		16 (24)	10 (15)	
	IV	4 (6)	1 (2)		2 (3)	3 (5)		4 (6)	1 (2)		2 (3)	3 (5)		4 (6)	1 (2)	

Results are expressed as number (percentage). NS: not statistically significant (*p* > 0.05). *MYLK*
^1^ (NM_053026); *ADH1B*
^2^ (NM_000668).

## Data Availability

The data presented in this study are available on request from the corresponding author.

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
