# Peer review of "Dysregulated Expression of Three Genes in Colorectal Cancer Stratifies Patients into Three Risk Groups"

_cancers, 2022, doi:10.3390/cancers14174076_

Round 1

Reviewer 1 Report

Major comments:
The proposed study is based on the supervised analysis of pre-selected 28 genes from previous study. There is no enough background on the selection of the 28 genes or appropriate citation.
Suggestion: This study relies on these pre-selected 28 genes and background must be addressed clearly.
Specific comment 1: Authors has mentioned (GSE39582; n =562) but actual sample size in this study has more samples. The mentioned study has total 585 samples out of which 443 samples were utilized for unsupervised analysis, 123 as validation datasets and 19 were non tumoral.
Suggestions: The reasoning behind selecting 562 samples out of 566 tumoral samples must be explained.
Specific comment 2: Authors only used 1 independent dataset for the validation. This is not sufficient.
Suggestions: A quick GEO database search showed there are many independent studied in CRC. It is recommended to validate in at least two independent datasets or strong reasoning must be provided for validating with one independent dataset.
General comment:
1. The KP survival plots will be more readable if you can change the X- axis in the interval of 12 months.
2. Suggestions for color codes in Figure 6 for KP plots would be red, blue and black. Some colors are not very reader friendly.
3. The term “The global transcription profile” is mentioned several times. It is recommended to describe the term.

Author Response

ANSWER TO THE COMMENTS OF THE REVIEWERS

REVIEWER 1:

Comment 1.-

Major comments:

The proposed study is based on the supervised analysis of pre-selected 28 genes from previous study. There is no enough background on the selection of the 28 genes or appropriate citation.

Suggestion: This study relies on these pre-selected 28 genes and background must be addressed clearly.

Specific comment 1: Authors has mentioned (GSE39582; n =562) but actual sample size in this study has more samples. The mentioned study has total 585 samples out of which 443 samples were utilized for unsupervised analysis, 123 as validation datasets and 19 were non tumoral.

Suggestions: The reasoning behind selecting 562 samples out of 566 tumoral samples must be explained.

Answer to comment 1.- Following the comment of the reviewer a new sentence has been included in the Results section (Sub-section: Validation of the prognostic score in two independent series of sCRC patients) of the revised version of the manuscript, describing the reason for removing 4 patients from the validation study.

Comment 2.- Specific comment 2: Authors only used 1 independent dataset for the validation. This is not sufficient.

Suggestions: A quick GEO database search showed there are many independent studied in CRC. It is recommended to validate in at least two independent datasets or strong reasoning must be provided for validating with one independent dataset.

Answer to comment 2.- According to the suggestion of the reviewer, a second validation study has been added from another GEO-independent dataset (Figure S2). In addition, a new (brief) sentence has been included in the Abstract, Material and methods and Results sections of the revised manuscript, in this regard.

Minor comments:

General comment:

  1. The KP survival plots will be more readable if you can change the X- axis in the interval of 12 months.

            The X-axis of all KP survival plots now displays information at 12-month intervals, including KP from supplementary material.

  1. Suggestions for color codes in Figure 6 for KP plots would be red, blue and black. Some colors are not very reader friendly.

            The colors in Figure 6 have been changed, following the reviewer's suggestion.

  1. The term “The global transcription profile” is mentioned several times. It is recommended to describe the term.

It is now clearly described the term “The global transcription profile” in both abstract and results sections of the revised manuscript.

Reviewer 2 Report

Sporadic colorectal câncer (sCRC) is one of the most frequent causes of cancer death in the western world and the disease was characterized by heterogeneous clinical manifestations and patient prognosis. The manuscript “Dysregulated expression of three genes in colorectal cancer stratifies patients into three risk groups” by Carreño, A. R. et al. analyses the differential expression of 28 genes and their possible association with tumour local aggressiveness and metastatic processes in 66 sCRC patients. The data analysis was carried out on the expression platform PAM50; nCounter; NanoString Technologies. The authors shows a strong association between the altered expression of three genes ADH1B, BST2 and FER1L4 and the clinical outcome of sCRC and allows patients to be stratified into three risk groups that are directly associated with different 5-year survival rates. The research was appropriate designed and the methods were adequate to respond the questions. The results were duly described and support the conclusions. In my opinion the manuscript deserves be publish in Cancers.

Author Response

REVIEWER 2:

Comment 1.- Sporadic colorectal câncer (sCRC) is one of the most frequent causes of cancer death in the western world and the disease was characterized by heterogeneous clinical manifestations and patient prognosis. The manuscript “Dysregulated expression of three genes in colorectal cancer stratifies patients into three risk groups” by Carreño, A. R. et al. analyses the differential expression of 28 genes and their possible association with tumour local aggressiveness and metastatic processes in 66 sCRC patients. The data analysis was carried out on the expression platform PAM50; nCounter; NanoString Technologies. The authors shows a strong association between the altered expression of three genes ADH1B, BST2 and FER1L4 and the clinical outcome of sCRC and allows patients to be stratified into three risk groups that are directly associated with different 5-year survival rates. The research was appropriate designed and the methods were adequate to respond the questions. The results were duly described and support the conclusions. In my opinion the manuscript deserves be publish in Cancers.

Answer to comment 1: We thank the reviewer for his positive comments about the manuscript and the work contained in it.

Reviewer 3 Report

In this manuscript, the authors have examined the differential expression of these 28 genes in 66 consecutive sCRC patients for any potential relationships to tumor local aggressiveness and metastatic processes. They demonstrate a significant correlation between the clinical outcome of sCRC patients and the altered expression of three genes, e ADH1B, BST2 and FER1L4. The manuscript is well written and of high clinical significance. However the authors should confirm the prognostic significance of the three genes by analyzing the respective proteins and their association with prognosis.

Author Response

REVIEWER 3:

Comment 1.- In this manuscript, the authors have examined the differential expression of these 28 genes in 66 consecutive sCRC patients for any potential relationships to tumor local aggressiveness and metastatic processes. They demonstrate a significant correlation between the clinical outcome of sCRC patients and the altered expression of three genes, e ADH1B, BST2 and FER1L4. The manuscript is well written and of high clinical significance. However the authors should confirm the prognostic significance of the three genes by analyzing the respective proteins and their association with prognosis.

Answer to comment 1: We fully agree with the reviewer that validation at the protein level would greatly improve the manuscript. Precisely, we have started this study, both in the primary tumor and in peripheral blood, in a large series of patients with CRC. However, we still do not have solid results from the study, which we hope will soon be published in the prestigious journal Cancers.

Round 2

Reviewer 1 Report

The revised version has addressed all the major comments.

Author Response

We thank the reviewer for comments on the manuscript that helped to improve it significantly.

Reviewer 3 Report

The authors should discuss the association between the proteins of interest and their prognostic value with appropriate references.

Author Response

Following the reviewer's comment, a new paragraph has been added at the end of the discussion section of the revised version of the manuscript, describing previous associations found in the literature on the levels of BST2, ADH1B and FER1L4 proteins detected in tumor tissue and plasma of patients with sCRC and with other types of neoplasms, and their prognostic value. Additional references in this regard were also added.
